# A Model for the Implementation of Lean Improvements in Healthcare Environments as Applied in a Primary Care Center

**DOI:** 10.3390/ijerph18062876

**Published:** 2021-03-11

**Authors:** Oscar Morell-Santandreu, Cristina Santandreu-Mascarell, Julio J. Garcia-Sabater

**Affiliations:** 1Universitat Politècnica of València (UPV), 46730 Grau de Gandia, Valencia, Spain; osmosan1@upvnet.upv.es; 2IGIC, DOE, Universitat Politècnica of València (UPV), 46730 Grau de Gandia, Valencia, Spain; 3ROGLE, DOE, Universitat Politècnica of València (UPV), 46022 Valencia, Valencia, Spain; jugarsa@omp.upv.es

**Keywords:** lean philosophy, healthcare, continuous improvement, lean healthcare, value stream mapping, Covid-19

## Abstract

Companies operate in a competitive and changing environment requiring increasingly effective and efficient management strategies. Lean is a proven philosophy in the industrial sector having helped companies to adapt to rapid market changes; to economic, technical, and social complexities; and to customer needs. For this reason, companies in the service sector are adopting Lean to improve their service management and to achieve economic, social, and environmental sustainability. This paper presents a model which uses Lean tools to facilitate the introduction of Lean in the management of primary care centers. The results show the implementation of Lean improved primary care center management, achieved stated objectives, and demonstrated faster adaptation to environmental needs and changes. The Lean philosophy developed and applied in the primary care center proved useful at a professional level facilitating developmental changes and prompting lasting improvements by developing a sustainable work culture.

## 1. Introduction

In 1986, the Spanish health system created an institutional framework that is still in place today and has allowed the needs and expectations of the population to be adequately structured. Although it is considered one of the most efficient in terms of accessibility and shows, in general, a high level of quality and health outcomes at a comparatively lower cost compared to other countries, the system persists in a state of vulnerability. Various factors explain this situation, but here reference is made to those considered most relevant [1]:The Spanish population has the longest life expectancy in the European Union, standing at 83.4 years in 2017, 2.5 years above the European average. This is largely due to advances in medical treatments, which have meant a notable reduction in mortality and an increase in longevity of four years in the last two decades. This entailed an increase in health expenditure to 2371 Euros per inhabitant, of which 71% was publicly funded and 29% privately funded. Specifically, Spanish health expenditure amounted to 8.9% of the Gross Domestic Product (GDP), though still lower when compared to the 9.8% average in the European Union.The National Health System (NHS) provides a broad level of coverage, reducing financial, social, and geographic barriers to accessing care.The level of decentralization of services is reflected in the 17 health systems that manage healthcare resources across the Autonomous Communities.Waiting lists in the NHS remain a persistent problem. At the end of 2019, 704,997 people were awaiting a date for surgery, 5% more than the previous year. Specifically, 1.5% of the Spanish population was on the waiting list for surgery and 6.5% for consultations and diagnostic tests.Primary care is a key part of health services. However, coordination between different levels of care remains an organizational challenge, especially in order to improve care for chronic diseases.A health system unprepared for rapid growth in the flow of patients, which would lead to increased costs from more human resources (labor), materials (equipment) and financial resources (income).

Recent research by García and Rodríguez [2,3] indicated the need for health reforms in different areas, one of which was management. Efficiency gains are important as management represents between 20–40% of public spending, so there was, or should have been, real interest in improving its operational standards. This indicated the Spanish health system needed to continue to research to improve and reduce its limitations.

However, emphasis is placed on the fact that during this study the Covid-19 pandemic emerged, allowing the observation of new limitations on the health system [4]. The targeting of many of these tools during the Covid-19 epidemic led to organizational innovations, which were likely years away from implementation, and in some cases, during the pandemic was the best time to implement them, as indicated by Bonet [1,5,6,7].

Table 1 below summarizes the general problems and limitations that the Spanish healthcare system was already facing and should have been improving, and also includes new issues resulting directly from Covid-19.

In this context, the Spanish health care system must seek effective strategies to improve care processes so all patients can receive quality care according to their needs. By making changes, these process improvement systems must improve outcomes over time, avoid queues, and optimize information flow to patients and families. In addition, they must contribute to the effective functioning of the group, structure, or process, thereby improving quality and preventing conflicts; they must include the tools, techniques, and skills to generate a climate of trust and empathetic, fluid, and honest communication. The health sector was tasked with demonstrating tangible results through which it could reach levels of effectiveness, efficiency, and quality, chiefly driven by the relative dissatisfaction of the citizens (patients) and the need to control the high costs and overutilization of services [8].

The literature review indicated health systems more and more worked to implement these changes. Recent research [9,10,11,12,13] highlighted the Lean philosophy as a strategy allowing companies in any sector to adapt to rapid changes in the market, in the economy, in technical and social complexity issues, and in the needs of customers as well as for the health sector to adapt to the demands discussed above. A bibliographic review found great diversity, for example, in the area of emergency care, one of the most complex and haphazard areas [12,13,14,15,16] where this philosophy was applied for process improvement with successful results. Other researchers, such as Guzmán [6], also showed how, in the last 10 years, general hospital departments published results of process improvement and time optimization in emergency medicine, surgery, pharmacy, nursing, oncology, and administration reflecting the success of having implemented the Lean philosophy. This involved adapting the different industrial management approaches to the healthcare field, seeking the efficiency and effectiveness of the different processes and productivity systems. At the healthcare level, these new adapted approaches are a challenge of innovation in the new management of the service sector [17,18]. This new way of working in the health care field is known as Lean Healthcare, which, according to Dahlgaard, et al. [19], seeks to develop a service standard marked by patient satisfaction via continuous improvement in the value-creating care delivery processes and activities, and to engage all participants in the value chain of care delivery work to reduce waste and to promote patient value creation throughout the patient flow.

Lean was born in the industrial sector during the 20th century and is considered a philosophy of work based on the participation of people that defines ways to improve and optimize a production process by focusing, identifying, and eliminating all unnecessary processes and/or activities, called “waste”, thus facilitating the stability, efficiency, and productivity of the organization [20,21].

The Lean philosophy concludes that adopting and implementing the different Lean principles in the organization generates a positive change with evident results in healthcare, such as [2,20,21]:Reduced waiting times.Improved patient satisfaction.Improved productivity.Reduced costs.Service capacity increase.Improved response time to patient/patient demands.Improved staff satisfaction.Reduced number of errors.

The implementation of Lean is based on knowledge of the techniques and tools that allow process and/or activity improvement leading to optimization of those processes or activities thereby minimizing times, increasing profitability, adding value, etc., in short, improving client satisfaction. The techniques and tools used in Lean implementation are divided into three groups (based on “The Toyota Production System House”), although more can be found [6,19]:

Among the different techniques and tools that help to implement Lean there are several different classifications [10,20]. For example, the one based on the “Toyota Production System House” presents three types of tools for the different phases of Lean:Diagnostic tools: Value Stream Mapping (VSM).Operational tools: 5S, Single-Minute Exchange of Dies (SMED), Total Productive Maintenance/Team approach to problem solving (TPM), Kanban.Tracking tools: Visual Management (VM), Key Performance Indicator (KPI).

These and other authors [10,20,22], have also indicated that the application of each tool depends on the real situation in which the organization or company finds itself, as experts in the field do not agree when it comes to identifying them, classifying them, and proposing their scope of application. These tools can be used independently or in conjunction with other tools (e.g., process diagram, Ishikawa diagram, spaghetti diagram) not necessarily related to production process improvement in themselves, but that are equally useful for the development of the Lean philosophy and that serve the continuous improvement of the organization.

Before implementing a process, a diagnosis must be made of the current or starting situation in which the organization or company finds itself. Once the current situation has been identified, the ideal to be achieved must be visualized, which is done by analyzing the value flow.

One diagnostic tool, VSM, has evolved as fundamental in healthcare environments. Although not the only tool available for the diagnostic phase, VSM’s inherent visual nature makes it well suited to and more commonly used in healthcare [10,18,20,22,23,24,25]. It is carried out by the illustration of both the flow of material and the flow of information from the moment the customer places an order or requests a service until the delivery of the product or the service [26,27]. At the same time, it helps the process of continuous improvement [28], which is one of the objectives of Lean, and helps to maintain sustainability (social, economic, environmental) wholly or partially in organizations or companies. Elshennawy et al. [29] indicated there was a high level of awareness of Lean tools using the Lean Sustainability Assessment Framework (LSAF). They found approximately 80% of hospital managers were aware of various Lean tools, such as: 5 s, continuous improvement, waste disposal, 5 whys, waste types, and VSM.

VSM, as a methodology proposed in research [20], is part of the planning phase of Lean implementation and must be developed for each product family, determining what adds value for the customer (value), taking into account the process flow (stream), and creating its visual documentation (mapping). Consequently, in order to determine whether activities add value or not, prior knowledge of the tool is required along with involvement on the part of individuals and managers. This entails considerable investment in both people (training) and time. The preparation process is complex and must be carried out correctly and with great precision in order to be effective. Prior to the VSM phase, this research presents the phase of diagnosis and training, in line with which it recommends the use of tools (even if they are not directly associated with Lean) that will allow the representation of processes as well as the identification of inefficiencies and, as a result, pave the way for an eventual and appropriate application of VSM.

One of the tools that accompanies VSM, facilitates its application, and allows the determination of a generalized view of the process operations is the process diagram [30,31], providing insight into the service and identifying the different tasks and stages of the process in its entirety. Another tool that can be used is the Ishikawa diagram, which allows the origin of process failures to be identified. Therefore, starting with the use of a wide set of tools that facilitate Lean helps to pinpoint which activities do not add value and can therefore lead to their reduction and/or elimination thereby giving rise to one or more improvements in the process. This tool allows the creation of a common and standardized language within the company, which improves the effectiveness of the processes and the staff [20,32], and once part of the corporate culture, it allows maintenance of economic and social sustainability [25,33,34].

Continuing along these lines, the research focused on diagnosing, analyzing, and improving processes within a health center.

Health Centers are a fundamental pillar within the Spanish health system. The Health Center is the physical and functional structure in which the Primary Care Team works and constitutes the basic structure of the NHS [10]. The centers carry out their functions and constitute the first level of contact with the community, the family, and the individual as a patient of the NHS allowing the delivery of activities of promotion, health prevention, diagnosis, treatment, and social reintegration. The care provided must meet the criteria of accessibility and continuity of services, as well as coordination with other levels of care both health (specialized and hospital) and social (activities for the community).

Given that current healthcare delivery models were complex organizations, they required dynamic transformation through the implementation of innovative organizational measures to adapt to the changing needs of the environment and the population, to the available resources, thereby increasing competitiveness [35]. This implied that organizational management should promote a culture of change through the training of teams oriented to the strategic commitment of the company (analyze, identify, understand the flow of information and customer demand, and encourage the participation of staff in the overall process) [36,37]. This culture change should in turn generate added value for the final product (patient), eliminate those things that do not add value, promote continuous improvement, and ensure sustainability [38,39,40].

In this context, the research question to be addressed was whether it was possible to use the Lean philosophy by training one of the medical managers of a primary care center and, together with his leadership, redesigning the care processes with the aim of improving patient care.

In order to answer the research question and resolve the limitations of a health care system whose objective it was to improve patient care, this research proposes as a positive contribution a methodology that allows improvement in the health care processes in a health center using the principles of the Lean philosophy and the action research methodology. The methodology can be replicated in other centers that want to embark on continuous improvement by facilitating the process. In addition, in the academic field, this research expanded the conclusions of recent work [22], which indicated the lack of research with this approach in the health sector and further indicated that those existing were isolated and without solid evidence of sustainability over time.

This document consists of six parts. After the introduction above, the second part presents the proposed methodology. The third and fourth parts describe the practical application of the methodology in the health center in the first quarter of 2020 (without Covid-19) and the second quarter of 2020 (with Covid-19). The fifth part of this paper presents the discussion of the primary results obtained along with some conclusions. Finally, the sixth part presents the limitations of the study and some future lines of research that would give continuity to the study.

## 2. Materials and Methods: Model for the Implementation of Process Improvement in a Healthcare Environment

As mentioned in the introduction, the industrial sector has been innovating for many years with regards to various dynamics of organizational management, which have shown in a clear way the improvement of product manufacturing and service provision, as well as increased consumer satisfaction [41,42,43]. These organizational improvements are important because they make the organization more flexible, efficient, transparent, and focused on understanding and meeting consumer needs. That is why the service sector, including the health system, and public administrations in general, are now adapting and implementing some of these improvements.

This study focused on tools that help to change entire processes, value chain mapping [44], which in turn can be supported by the process diagram and the Ishikawa diagram (both effective tools for analyzing the different causes of a problem). Their main advantage is that they make it possible to visualize the different cause-and-effect chains, facilitating subsequent analysis to evaluate the degree of contribution made by each of these causes [45,46].

Following the principles of the Lean philosophy [47,48] and using action research methodology [18], this section presents a scientific methodology as shown in Figure 1 below that allows the redesign of processes and that can be replicated with the same purpose in other health centers in order to improve patient care.

Previous research similar to this paper [18] suggests that there are not many references in the health care field to the action research approach [49,50,51]; it is a sector more focused on case studies.

However, if we take consider the characteristics and requirements of action research, its use in this research is justified.

Action-research methodology is characterized by a collaboration between the researcher and a member of the organization in question in order to solve the identified problems [52,53]. It presents a collaborative approach in both the diagnosis of the problem and in the development of its solution. In addition, this type of study assumes that the environment is constantly changing with both the researcher and the research being part of that change [54,55]. Although action research can be divided into categories: Positivist, interpretive, and critical, in this paper we focus primarily on the critical category because it corresponds to research that adopts a critical approach to processes and seeks direct improvements.

The methodology is suitable for this type of research because, in addition to meeting its main characteristics (mentioned above), it fulfills its three requirements [54,56]:It is applied to improve specific practices. Action research is based on action, evaluation, and critical analysis of practices based on data collected to introduce improvements in relevant areas. It enables the creation of knowledge.It is facilitated by the participation and collaboration of a number of people with a common purpose. It requires close collaboration between the researchers and the organization.It focuses on specific situations and their context. Results are obtained from the efficiency of the process carried out.

There are five phases of action research, which are constantly repeated giving rise to possible cycles [18]:Plan to initiate the change.Implement the change (act), observe the implementation process and its consequences.Collect and review data.Reflect and evaluate.Readjustment.

There are some authors [57] who recognize that these stages can overlap, allowing the organization to continuously adapt to the environment and not allowing the initial plan to become obsolete.

Furthermore, they construct a flexible process linked to continuous improvement whereby each step results in a better and more appropriate outcome that is in line with that desired and ends up creating value for the customer and improving the job satisfaction of the organization’s staff.

Other authors [58] represent the different phases of action research by adding a preliminary phase and a final phase to Lewin’s original model [1,59]. This gives rise to cycles that come to an end only if the stakeholders have solved all the identified problems.

Like all research methodologies, it has advantages and disadvantages. The main advantage of the action research approach lies in the ability to analyze the phenomenon in greater depth each time, resulting in a deeper understanding of the problem. The main disadvantage is that it assumes that each process takes a long time to complete, which is not always the case. The main purpose of using action research is to seek the participation and involvement of the researcher as an agent of change along with the other members of the organization because, in the health sector, managing change requires the participation of the entire team involved in the processes identified for improvement [49]. This will allow the knowledge acquired to have a cyclical effect, which in turn will allow further improvement and sharing of that improvement with other organizations and researchers [18].


*Phase 0: Preliminary: Identification of the Team and the Patient Family to be Improved*


Firstly, the implementation of Lean philosophy and action research methodology requires the formation of a team where the majority of the workers are involved to the extent possible (depending on the size of the organization), or at least the participation of representatives of all the areas/departments that constitute the organization [48].

For the proper functioning of the team, two relevant roles are established within:The improvement promoter: A person within the organization with the capacity (formal power) to provide resources and impose changes as necessary [60].The facilitator: A person who must have the quality to know and validate the tools to be followed and at the same time must understand the needs of the organization. This will allow the process to be followed and fulfilled with the necessary rigor [46].

In addition, a representative of each role that may be affected by any of the possible improvement changes should be part of the team.

At the same time, the patient family to be improved should be selected [44]. Although in the process of defining the team it is already assumed that a specific group will be improved, only one family of patients in particular should be analyzed, as, for instance, the problems of a primary care center for adults are not the same as one for children, nor are the cardiology and traumatology consultations in a specialty center, although some improvements can be extrapolated. Based on the selection of the patient family, the team can be completed or modified ensuring that all parties are represented.


*Phase 1: Planning*


Subphase 1.1: Initial Analysis to Identify Processes and Objectives

Once the team has been established and the patient family identified, this phase describes the processes found within the flow of the same family, identifying all the processes that are involved in that family, and observing the existing interconnections. Additionally, the phase establishes how much information is available in a descriptive way.

The aim of this phase is to establish an overview of what is being done, how it is being done, and who is doing it. This overview allows the establishment of the general objectives and in connection with subsequent phases identifies where changes are required which will lead to possible improvements.

For this purpose, analysis should be carried out in line with expert research and publications [20] to facilitate selection of the tools and/or techniques that help in the implementation of Lean philosophy and that best suit the specific situation and characteristics of the organization in question.

It is important to emphasize that in this phase it is neither mandatory nor recommended to have a detailed vision of what is being analyzed. Instead, all team members can share a global vision of the processes to be improved and, if necessary, of the adjacent processes that will be affected.

Subphase 1.2: Definition of Improvement Indicators and Target Setting

Once the improvements to be made have been identified, they are established as objectives to be achieved. These objectives must be based on indicators. The indicator is a key aspect in any improvement situation [46,61], which will allow the determination of objective achievement without excessive cost.

When setting objectives, it is advisable to follow the SMART rule [62], i.e., they should be:Specific: Number or percentage that avoids generality.Measurable: Number or series of numbers that allow the measurement of its effectiveness.Achievable: Objective is achievable.Realistic: Objective is realistic.Time-limited: There must be a timeline for when it must be met.

Subphase 1.3: Definition and Analysis of the Current State (As-Is)

In any process of continuous improvement, it is necessary to find out in detail what is happening and why it is happening, and to identify the main problems faced; what affects the objectives previously set and which ones do we want to achieve? In this phase, the starting situation must be identified, determined, and considered. It is important that the whole team perfectly understands the problems presented by each section and the existing improvement opportunities, regardless of the role of each team member.

In this identification phase, different organizational management tools and/or techniques based on Lean principles can be applied, such as process diagram [30,31] and VSM [44], IS/NO matrix for the definition of problems, the spaghetti diagram, the Ishikawa diagram [63], 5 whys, or any other available tools for the correct identification and analysis of problems and/or opportunities [64].

In this phase, it is important to select tools that are not cumbersome and are easily understood by the team.

Subphase 1.4: Definition of the Ideal Future State of the Process to be Improved (To-Be)

Once the current situation has been analyzed and the achievement indicators have been established, there is then sufficient and relevant information to represent the future state, keeping in mind that changes and results in continuous improvement are rather long term [44]. It is during this phase that the representation of the future process deemed necessary is carried out, and it should be borne in mind that the team or facilitator must again select the best tool to carry out this phase.

Fundamental to this phase is for the teams to avoid partial improvements in order to improve the work of select professionals, but rather to maintain the focus on the ultimate improvement patients will see.


*Phase 2. Implementation*


Once the current situation has been assessed and actions have been established, it is possible to establish the action plan along with the steps to be completed.

The objective of this phase is to determine the actions to be carried out within the individual processes as well as to decide how to start working with those new processes.

All staff within the organization identified to make process improvements must be clear about how the new processes are to be carried out, how the tasks are determined, and the interconnections between them.

It is essential in this phase that all actions have someone responsible for monitoring, which is routine in hospital environments and is usually left to the team in general as and when required. The person responsible for each action must record the reasons for success or failure of the action in order to redirect focus (Phase 5: Readjustment) or to finalize the improvement process (Phase 6: Completion) and notify the team what they consider the key components of success or failure of the actions.


*Phase 3. Verification of Results*


Like all improvements and plans, it is necessary to check progress. The aim of this phase is to check the action results achieved. Therefore, data must be collected in accordance with the established indicators.


*Phase 4: Consideration and Evaluation*


In this phase, the primary activities to complete are as follows:Analyze the project closing with the team.Establish future improvements based on the actions not implemented and the objectives not reached.Summarize the lessons learned in this process for future processes.

At the end of this phase the whole team must be aware of the actions executed and those that fell short, and the objectives achieved and those that came up short, and the degree of achievement.


*Phase 5: Readjustment*


The readjustment phase consists of reviewing and evaluating those objectives achieved and those not achieved, and the causes of each. Care is taken to focus on those objectives that have not been achieved and modifying the action plan and its activities accordingly.


*Phase 6: Completion*


This phase is only reached when, as indicated above, all identified problems have been resolved by the stakeholders.

Section 3 and Section 4 present the results of the application of this methodology in two scenarios corresponding to two time periods:From Q4 2019 to Q1 2020.From Q1 2020 to Q2 2020 (Covid-19).

These scenarios were not planned, but rather were the result of the unexpected and unforeseen situation arising from Covid-19.

At the end of 2019, one of the authors (who was also a physician at the primary care center in question) decided to apply knowledge they had gained regarding the Lean philosophy for process improvement at their workplace. There were two time-based objective sets as follows:Q4 2019:
Create a working group to analyze the processes of the primary care center.Identify opportunities for improvement.Establish an action plan with its corresponding actions.Establish indicators to measure the results.Implement the actions.
2020 and forward:
Measure the results on a quarterly basis.Adjust the plan to establish a system allowing continuous and sustainable quality improvement [38,39].


This required a process that allowed the transformation of the organization and not only implemented adjusted tools and/or practices that helped Lean [35], but that also corresponded to the action research meta-logic as shown in Figure 2 below.

However, at the beginning of the process, and just at the point of collecting the first set of data (first quarter of 2019), Covid-19 appeared, changing the whole scenario and characteristics of the healthcare environment.

The research team and collaborators/participants who were working on the improvement of processes in the primary care center realized that a work culture had been created in a short time [18,35]. Moreover, they decided to continue with the methodology they had established to adapt to the new needs of the environment as existing in-depth knowledge of the center’s processes allowed its staff to more quickly identify which activities were necessary and which were not according to the new needs of their patients.

## 3. Application of the Proposed Model to a Primary Care Center for the Improvement of Patient Care

This section presents the phases of the model proposed in the previous section corresponding to the first scenario.


*Phase 0: Preliminary: Identification of the Team and the Patient Family to be Improved.*


Following the model proposed in the first phase, the team and the patient’s family were identified within the Primary Care Health Center selected, which consisted of a portfolio of services distributed at different levels serving a population of 15,527 people.

To identify the people who should be part of the team and who should occupy the roles of promoter and facilitator, the organizational chart of the Health Center was analyzed, which in this case was a health center with a very basic structure, as shown below in Figure 3:

This organizational chart shows the Health Center had a simple hierarchical structure with three main levels: The Department Management, the Primary Care Medical Direction, and the Primary Care Medical Team (Medical Director of Basic Health Zone, Nursing Coordinator, and the Primary Care Team, all the workers in the center) working in an organization seeking participation and collaboration).

To establish the best team, the roles, tasks, and responsibilities of all the workers who made up the primary care team were analyzed. This gave rise to extensive tables (which contained much more than the pure functions described in contracts and were reviewed with the members of each team before correcting as necessary), which were considered inappropriate to incorporate because of their extensive nature. However, it is relevant to mention this task as the information collected from it helped structure the team and establish the roles of this phase, as shown below in Table 2.

Therefore, in this case, the facilitator led the team. The facilitator was one of the doctors of the center who held the position of medical director of the basic health area and who had trained in improvement methodologies for several years. They were in charge of creating a new organizational culture by promoting change through knowledge of the functions of each member of the primary care team and their interrelationships. The size of the Health Center did not require active incorporation of the rest of the staff into the team, but their concerns and needs as passive subjects within the organization were still considered relevant.


*Phase 1: Planning*


Subphase 1.1: Initial Analysis to Identify Processes and Objectives

Once the family was identified and the team was formed, the initial situation was analyzed.

In order to do this the most appropriate tool had to be chosen. As discussed in the introduction one of the most appropriate tools was considered to be VSM. However, it was decided best to start with a process diagram, as shown in Figure 4 below. This allowed an initial identification of the activities that made up the operating process of the primary care center.

In this As-Is process diagram, generated in the last quarter of 2019, the health care process activities of the Health Center during a 7-h working day are presented. The diagram shows a hierarchical distribution in three levels of care including administration, physicians, and nursing (nursing and auxiliary nursing technicians), through which the patient passes according to their care needs. The auxiliary nursing technicians were not included in this phase as their main function was logistical, which did not normally intersect with the care process.

In this diagram, it was necessary to highlight the differentiation made according to whether the assistance demanded by the patient was or was not a programmed activity. If the need was for an ordinary or scheduled appointment, the patient went from the access area to the doctor’s office to resolve the problem because they had specific opening hours. On the other hand, if the demand was unscheduled, it was necessary to differentiate whether or not there was a vital risk for care. If there was a vital risk, the doctor in charge of the emergency care was notified of the emergency, and they would leave their scheduled activity and go to the auxiliary consultation, while the nursing staff performed specific triage to classify the emergency care reason.

After the initial analysis, this phase ended with the establishment of the following objectives as priorities:O1: Improve the quality of patient care at the Health Center.O2: Improve worker satisfaction as indicated.O3: Improve professional satisfaction.

However, it was decided the priority target on which to initially focus was O1, improve the quality of patient care at the Health Center. Within this objective, which entailed many management tasks, the focus was on “calendar management”, because this activity produced the greatest work overload and most damaged the quality of care and patient satisfaction. Therefore, it also affected the other two objectives, O2 and O3. Following the Lean philosophy allowed improvement in the process management without damaging, and perhaps even improving, the care and quality results.

Subphase1.2: Definition of Improvement Indicators and Target Setting.

This phase aimed to establish the indicators to mark the success or failure of the planned measures. The task was performed by brainstorming and then selecting indicators that were calculable with the available data. The indicators were set as follows:Number of emergency visits: Represents the number of emergency or walk-in visits that arrive and that are scheduled at health centers in a given period of time.Number of scheduled visits: Represents the number of demand appointments scheduled in a given period of time.Number of telephone appointments for emergencies.Number of telephone appointments for a scheduled activity.Number of workers providing direct patient care service.Delayed care: Number of business days until a patient can make a scheduled appointment.

Once the indicators were established, the Health Department Manager validated them as they held ultimate responsibility for the proper functioning of the Health Center.

These indicators were extracted from an internal database management tool of the health services of the autonomous community in the period from October to December 2019. Table 3 below presents these indicators:

In addition, it is necessary to state the delay of required appointments to the health centers was established by official bodies as being within 2 days.

Subphase 1.3: Definition and Analysis of the Current State (As-Is)

After the analysis in Subphase 1.1 via the As-Is diagram and the setting of objectives improvement, which was calendar management, the detailed analysis of the different aspects of improvement was performed. This was done in two steps.

In the first step, the analysis of the actual situation in which the Health Center worked in relation to “calendar management” was completed and the premises of general improvements on which to work were identified. For this task and due to the situation, the Ishikawa diagram as shown below in Figure 5 was considered the most appropriate tool.

The Ishikawa diagram shown in Figure 5 below was created, and through brainstorming, the objective to improve “calendar management” was defined allowing visualization of the critical or most vulnerable points subject to possible changes in relation to the activity of the Health Center.

Instead of those categories used classically such as Man, Machine, Materials, Method, Environment (4ME), it was decided to group the branches into the following healthcare-focused categories:Patient-dependent factors.Primary Care Team-dependent factors.Factors dependent on the organization of the center.Factors dependent on the structure of the center.Factors dependent on the laws that regulate the National Health System.

Figure 5 above shows there are inefficiencies in four of the five branches represented, as the factors depending on the laws of the National Health System were static. These inefficiencies are marked with a red star on Figure 5.

In the second stage, an analysis of the calendar functionality was carried out as shown in Figure 6 below in order to visualize the critical points of the particular process. The Health Center worked with a calendar of 35 ordinary plus scheduled daily appointments, so taking into account the doctor’s working day was 7 h with a half hour break, this left approximately 11 min for each patient. However, the doctors not only provided service for ordinary plus scheduled appointments, they also attended to unplanned appointments, which can be classed as urgent or non-urgent. In reality, this means that the doctor would attend to the patients for approximately 50 appointments, of which 13 were not planned and had to be inserted into the daily schedule without following any established protocol. This meant that the time established per patient was reduced from 11 min to 7.8 min (3.2 min less). In fact, it was discovered the time for patient care was less than 7.8 min due to continuous interruptions in terms of unscheduled demand generated by the schedules. In addition, a period of time was set aside from 1400h to 1425h for doctors to carry out administrative tasks related to clinical records such as prescriptions, reviewing status of patient history, conducting consultations with specialists, requesting complementary tests, etc.

In order to strengthen the improvements, it was necessary to differentiate the types of assistance given.

Any care provided by a health professional to the population of the Health Center was considered direct care. Direct care, whose distribution is shown below in Table 4, consisted of offsite or telephone assistance, which represented 20% while the rest (80%) was practically onsite.

Subphase 1.4: Definition of the Ideal Future State of the Process to be Improved (To-Be)

The objective of this phase was to design the future process of how to work to reduce inefficiencies and to achieve the proposed goals. With the information gathered through the exposure of the problems detected in the previous phases a future diagram was proposed allowing the reorganization of the care activity of the center and therefore the management of the calendars, the result of which is shown in Figure 7 below. Use of the tools with which the team were already somewhat familiar continued due to their previous effective use during the first quarter of 2020, namely the process diagram and the Ishikawa diagram.

The To-Be-2020 process diagram (Figure 7) shows the hierarchy of patient care was maintained, but a fourth level was added after the separation of nursing and auxiliary nursing technicians’ functions, which allowed for organizational and structural differences in terms of the patient demands. In this case, the activities that underwent modification are marked with blue shading.


*Phase 2: Implementation*


In order to initiate the changes mentioned above, in this phase, the following action items and the activities to be executed were defined. The facilitator designed the implementation plan for these actions, which were then executed in the first quarter of 2020 as shown in Table 5 below.


*Phase 3:*
*Verification of Results*


The action plan was carried out during the first quarter of 2020 and data collection took place at the end of the year. Table 6 below shows the results of the indicators used as a reference to evaluate the improvement results.


*Phase 4: Consideration and Evaluation*


For this phase, the work team met to reflect on and evaluate the results obtained. First of all, the indicator results shown in Table 6 were compared with those from the last quarter of 2019 (Table 3). The comparison revealed the following:The number of visits decreased by approximately 4.6%, from 66,604 to 63,563. Of a total of 63,563 visits, approximately 80% were programmed visits.The number of scheduled appointments increased by 9.62%, while the number of visits dropped by only 4.6%.The number of unscheduled appointments reduced by 57%. These data were relevant when compared with the total number of views, which were virtually the same as in the last quarter of 2019.The emergencies attended were approximately 15% of the total visits.The delay, whose ideal standard of 2 days was virtually achieved, having dropped from 2.46 to 2.02 days.

The conclusion was that the actions carried out have led to improvements as shown by the indicators.

Secondly, the improvements due to each action were evaluated. To do so, a brainstorming process was employed, the results of which are summarized in Table 7 below.

The conclusions reached in this phase were:The work carried out had a positive impact as shown by the indicators and the various improvements achieved.The actions resulted in improved patient care, but one quarter was not considered enough time for definitive conclusions. In addition, a tool should be developed to collect customer satisfaction data with respect to the actions carried out.The staff noticed a lessening of their workload and therefore have been able to perform better, but a tool is also needed to collect job satisfaction data.Table 7 above was created from the information gathered by those responsible for monitoring, as well as, to a certain extent, from the impressions conveyed by the staff. Therefore, there was a need to establish how to collect this information and make it uniform, easier to collect, and measurable.


*Phase 5: Readjustment*


For this phase it was considered too early to make readjustments. Given that the existing actions carried out brought positive results and in the evaluation phase improvements were proposed, it was decided the best course of action was to continue with the existing action plan and monitor for further indicator improvements while, at the same time, implement any indicated improvements.

However, the new health situation caused by Covid-19 made the team decide to reapply this methodology while taking into account the new needs. They realized that the best way to adapt to the changes was to know in depth the internal processes and improve them (see Section 4), to wit, they considered the previous work an advantageous starting point.

In a relatively short time, a new work culture based on the Lean philosophy was being implemented with which the staff felt comfortable even though for the time being it meant more work.

The conclusion of this phase was, starting from the current work situation in Q1 2020, to adapt the actions already being carried out and apply them to the new needs. Thus, a new cycle of the action research methodology was initiated.


*Phase 6: Completion*


As mentioned in Section 2 above, this phase is not reached until all the problems identified have been solved. In the previous phase, nothing was considered closed. They must be adapted to the new needs required by patients with Covid-19 as discussed in the section that follows (Section 4).

## 4. Application of the Proposed Model to the Covid-19 Situation

At the end of January 2020, the World Health Organization (WHO) declared an international emergency setting guidelines for balancing the demands of direct responses to the coronavirus [65,66,67]. The third Covid-19 case in Spain was reported on 3 March, and by the end of the March, the number of people infected reached over 94,000. In this short period of time, 8189 people died, 49,243 remained in a hospital, and almost 20,000 were discharged [68]. On March 14, 2020, a state of alarm was declared in Spain with a call for the population to be confined to their homes in order to stop the progression of the epidemic, to slow down the contagion curve, and thus to avoid the collapse of the health services, especially hospitals [68,69].

The methodology phases adapted to the Covid-19 pandemic situation are presented below.


*Phase 0: Preliminary: Identification of the Team and the Patient Family to be Improved*


Due to the concurrent new application of the Lean philosophy and the emergence of the scenario faced by Spanish healthcare with the Covid-19 pandemic, a decision was made to keep the same team as well as to continue with the same work model and the same tools as used previously (i.e., as shown in the process diagram and Ishikawa diagram) in order to face and adapt to this new scenario. The main reason for this decision was that the management team realized that applying this philosophy had allowed the Primary Care Team to become much more familiar with the work processes of their Health Center and they were more prepared for new needs being prepared to participate and collaborate on how to adapt the work process. An improved work culture had been generated in a short period of time.


*Phase 1: Planning*


Subphase 1.1: Initial Analysis to Identify Processes and Objectives

In view of the new Covid-19 situation, health care needed at both primary care and hospital level changed radically and required a restructuring of services and new care protocols adapted to the changing and uncertain environment.

This phase was not carried out as such because it was possible to take as a starting point the situation from the first quarter of 2020 as shown in Figure 7 (2020 To-Be process diagram), a situation considered already analyzed because it had been actively monitored.

However, the priority objectives in the current situation were modified as follows:O1-Covid-19: Maintain accessibility to the health system for the Health Center’s reference population.O2-Covid-19: Ensure, as far as possible, the safety of care consultations for both healthcare professionals and for patients.O3-Covid-19: Improve information between the different levels of care, giving priority and resources to primary care centers, which were the first level of health care.

However, to achieve these objectives, the team decided to prioritize objective O1-Covid-19 with the following assumptions:Maintain the number of non-Covid-19 appointments (face-to-face or telematic).Maintain the number of face-to-face services at below 30%.Maintain an average delay of less than 3 days.

This led to the decision to refocus on calendar management.

Subphase 1.2: Definition of Improvement Indicators and Target Setting

Due to the new situation and its uncertainty along with the total lack of forecasts faced by the Health Center, it was decided to adapt the indicators for monitoring improvements as follows:Number of telephone appointments in scheduled activity.Number of face-to-face appointments in scheduled activity.Number of unscheduled urgent appointments.Number of appointments in Covid-19 calendar.Total visits.Delay.

Clearly, it was not possible to have previous records of these indicators in the Covid-19 situation.

Subphase 1.3: Definition and Analysis of the Current State (As-Is)

In the new situation of the Health Center, it was not possible to carry out this phase. The 2020 To-Be process diagram (Figure 7) was taken as the current state, and this was the initial working model for this phase being considered as an As-Is diagram.

Subphase 1.4: Definition of the Ideal Future State of the Process to be Improved. (To-Be)

With all the information gathered and the experience gained, a process diagram was drawn up establishing the ideal future state in order to provide information on how to work to achieve the proposed objectives and to continue offering citizens quality and, above all, safe assistance in the face of the Covid-19 pandemic.

The main issue was to ensure that the system could continue to function. This purpose required creation of a channel solely for patients suspected of Covid-19 infection, with staff dedicated exclusively to this type of patient, as shown in the lower part of Figure 9 below.


*Phase 2. Implementation*


The action plan and activities to be implemented are shown in Table 8 below:


*Phase 3. Verification of Results*


Following the changes made to the processes to adapt to the Covid-19 pandemic, the indicators achieved during the second quarter of 2020 are shown below in Table 9.


*Phase 4. Consideration and Evaluation*


The indicators show that continuing with the Lean philosophy helped the Health Center to achieve the set objectives. During the second quarter of 2020, two of the three proposed objectives were achieved:The number of non-Covid-19 appointments was maintained, either by face-to-face or by telephone. In this period, 48,113 patients were seen, compared to 63,563 in the first quarter of 2020 and 66,604 in the last quarter of 2019.The number of face-to-face services was less than 30% at a value of 24%.

The delay of less than 3 days remained to be achieved.

In addition, the action plan and its activities resulted in the improvements shown in Table 10 below.

The five principal conclusions reached in this phase were:Considering the situation, the work carried out has had positive results, as shown by the indicators and the improvements achieved. However, the actions should be reviewed, mainly to reduce the delay in patient care.Due to the uncertainty of the situation as well as sudden changes caused by the peaks of health saturation, it was not possible to collect data concerned with patient satisfaction with care received, nor has it been possible to dedicate time to elaborate a tool to obtain this information as it is not the priority at this time.In terms of working environment, the staff suffer stress due to saturation, which is diminished when the new agenda is applied. Yet, this change is not enough. As is the case with patient data, it has not been possible to devote time to the development of a tool to collect data connected with job satisfaction.As in the previous phase, Table 10 was drawn up from information gathered by the managers and the impressions given by the staff. There is still a need to establish a method of collecting this information, making it uniform, easier to collect (i.e., not adding to the existing workload), and measurable. Again, it is not considered a priority at this time.The priority is to adjust the actions to reduce the delays in patient care, which have increased considerably, as shown by the indicators.


*Phase 5. Readjustment*


Due to workload and the unpredictability of peak saturation periods, the team considered that no readjustment should be conducted, that they should continue with the action plan shown in Table 8, and return to Phase 4 (Consideration and evaluation) after a continued period of Covid-19 stability. This decision was taken due to the fact that, during periods of epidemic outbreak, the delay in care cannot be taken into account as a reliable indicator.


*Phase 6. Completion*


The process cannot be considered complete and we must continue working and readjusting the action plans and their activities to achieve ever greater improvements in patient satisfaction.

For this reason, another cycle of the action research methodology has begun.

## 5. Discussion

The methodology developed and applied in the primary care center proved useful at the professional level to develop changes and achieve improvements. Therefore, it can be said that implementing the Lean philosophy in the healthcare sector helped to achieve the proposed objectives as described below.

As shown below in Table 11, the analysis of the processes in the way the health center worked and the changes made in the management of calendars improved the quality of care offered to patients as it managed to increase the average real time of care from 7.8 to 10 min. In addition, without delays in emergency cases, this also reduced the feeling of time-wasting walking from one place to another in the medical center and not belonging to a specific professional. For both employees and professionals, the possibility of planning and management led to a reduction in work stress and overload.

This initial implementation of the Lean philosophy in healthcare demonstrated process improvement and allowed the creation of an improved work culture within a team in which all participants were involved. The implementation made it possible to identify changes more quickly, to manage existing resources more efficiently, and to detect the need for new ones. The implementation aimed at increasing patient satisfaction and quality of care, and at the same time increasing the job satisfaction of the workers and professionals. However, with the arrival of the Covid-19 pandemic, the improvements advanced lost their efficiency and a similar, or even worse situation to the start, returned in terms of patient care and the overload of workers and professionals due to uncertainty and constant changes.

However, the new work methodology allowed for action in situations of vulnerability, such as the Covid-19 pandemic, thus allowing modification of work processes in record time.

After applying the model to both situations, it was found that the choice of team members with sufficient power to make changes was fundamental [46,60]. It was observed that as the selected staff had power and resources to make changes in the primary care center, these were delivered, but this was not the case with the changes required to de-bureaucratize throughout the rest of the hospital. This step was delayed until the Covid-19 situation when the hospital management pushed for changes in other sections of the center.

The use of indicators to set objectives, although not a novelty, had not been used in the center previously and proved to be fundamental [61]. Their use allowed the teams to focus on the analyses and improvements of specific actions instead of trying to make top-level changes at the departmental level, which would require more effort and resources with possibly less efficacy.

The use of existing along with routine and proven tools and techniques, such as VSM [34], BPMN, Ishikawa diagram, and process diagram proved effective and useful in this environment, despite the lack of experience with these tools within the team. However, the experience highlighted that strictness in the “impeccable” use of these tools only slowed down the process because they required previous training for a user to be truly proficient, which was not available at the time.

## 6. Conclusions

To conclude, three relevant conclusions of note were as follows:Implementing and adapting the Lean philosophy in the healthcare sector can help to make the necessary changes and improvements discussed in the introduction. In this particular case, identifying the need to reorganize the agendas according to the care needs mainly allowed the reduction of the worker care load, highlighted the need to create new patient care circuits according to the type of appointment thereby reducing waiting times for patients, and revealing how to better readjust human and material resources.The applicability of new Lean working methodologies makes it possible to be more sustainable in the long term in the different aspects covered by sustainability, which are summarized below:
Economic sustainability: Patient care times are reduced and improved and both material and human resources are managed more efficiently.Social sustainability: Internal work circuits are improved and the level of work stress is reduced due to the adaptation and continuous change of protocols that are adapted to the needs of the population and its environment. This has a positive impact on patients’ perception of the service provided.Environmental sustainability: Secondarily and after the establishment of specific protocols, exposure to biological risks, among others, can be minimized, contributing to environmental improvement.
The methodology proposed here can be extrapolated and used in other primary care centers to facilitate and reduce the barriers and limitations that are generated when working and introducing it for the first time.

## 7. Limitations and Future Lines of Research

This study encountered several unresolved or pending issues:Measuring the satisfaction of patients and workers at the Health Center. Still pending is the development of questionnaires to measure satisfaction, but the situation up to publication had not allowed it. Comments were based on the ad hoc feedback received from both the staff and the patients.As Lean implementation progresses, we might also consider the possibility of consolidating research with Six Sigma methodology, which facilitates the application of statistical methods and techniques for continuous improvement.The indicator times were established through observation of the calendars and follow-up with patients by the medical director and principally the nursing coordinator. Still pending is the development of a tool to collect information in a uniform, easy, and measurable way.At the quantification-of-results level, comparison with the post Covid-19 stage was virtually impossible due to the variability of the new processes, which were subject to constant changes, the declaration of the state of alarm, and their sustainability over time.Study the advantages and disadvantages of the various Lean tools as well as other alternatives in order to determine which of them are the most appropriate for the organization’s situation.Hospital management has considered replicating this methodology in other primary care centers and other areas in order to assess and improve their management and at the same time, to check whether the proposed methodology would serve as a checklist for the management being carried out.In terms of sustainability, the Lean philosophy contributes greater value to economic and social sustainability, having a much smaller impact on environmental sustainability.

All of the above areas are intended for further research by the team in the future.

## Figures and Tables

**Figure 1 ijerph-18-02876-f001:**
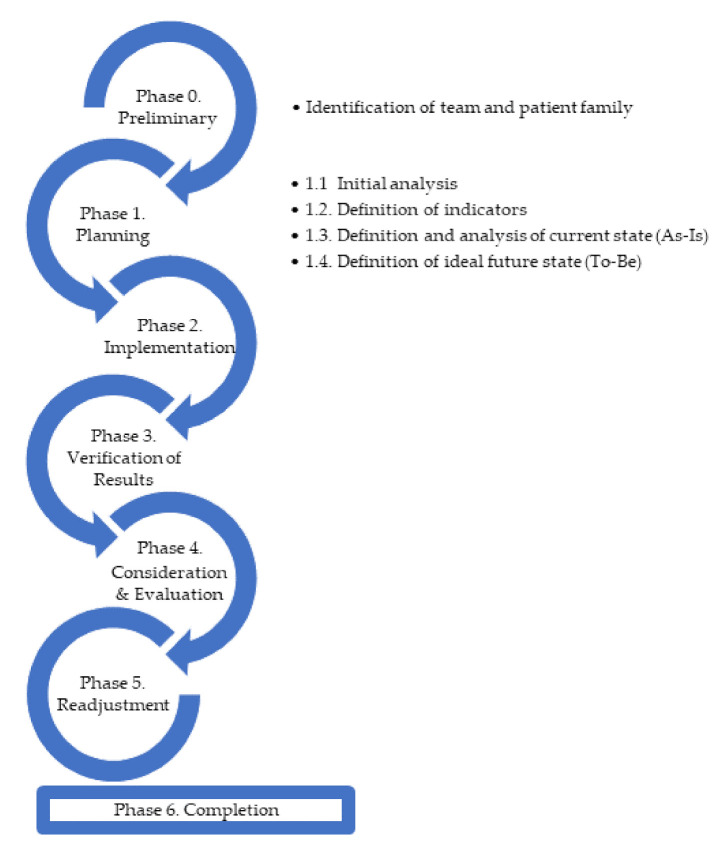
Phases of the Proposed Model.

**Figure 2 ijerph-18-02876-f002:**
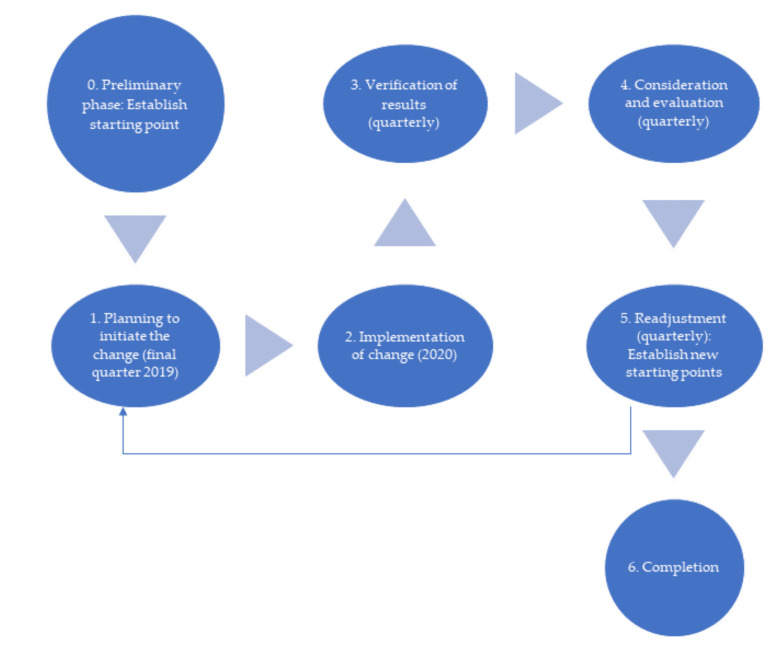
Action research methodology [58,59].

**Figure 3 ijerph-18-02876-f003:**
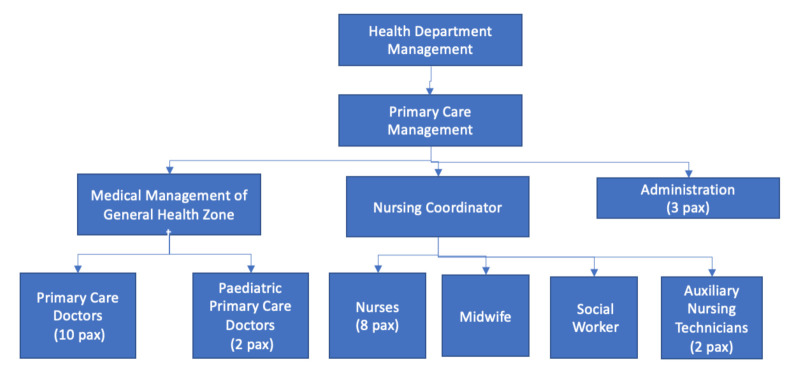
Health center organization chart.

**Figure 4 ijerph-18-02876-f004:**
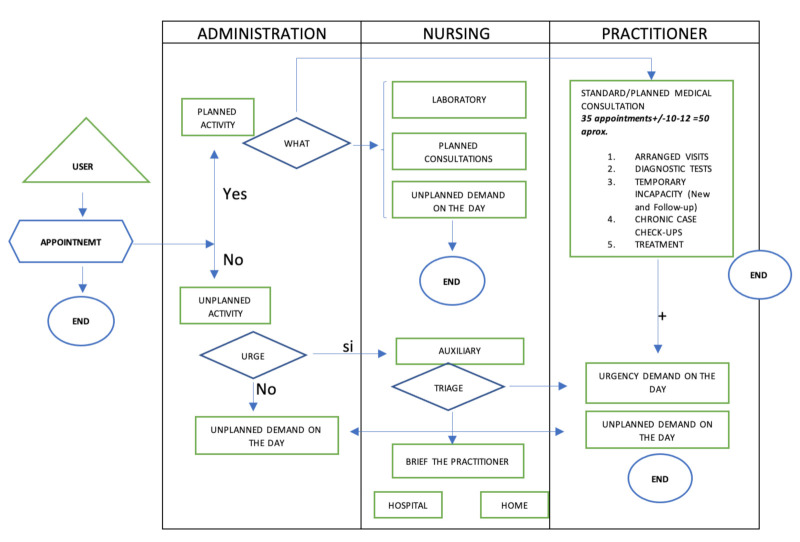
As-Is 2019 diagram of assistance processes.

**Figure 5 ijerph-18-02876-f005:**
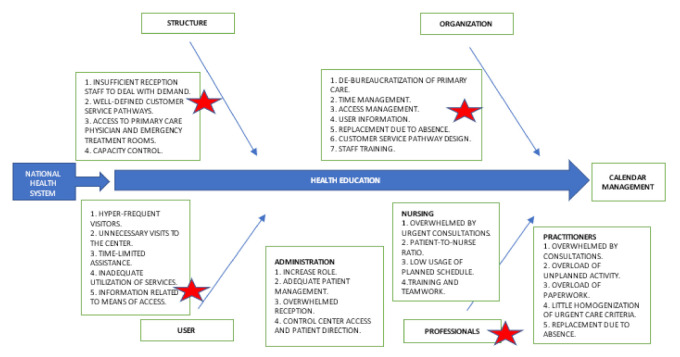
Ishikawa diagram.

**Figure 6 ijerph-18-02876-f006:**
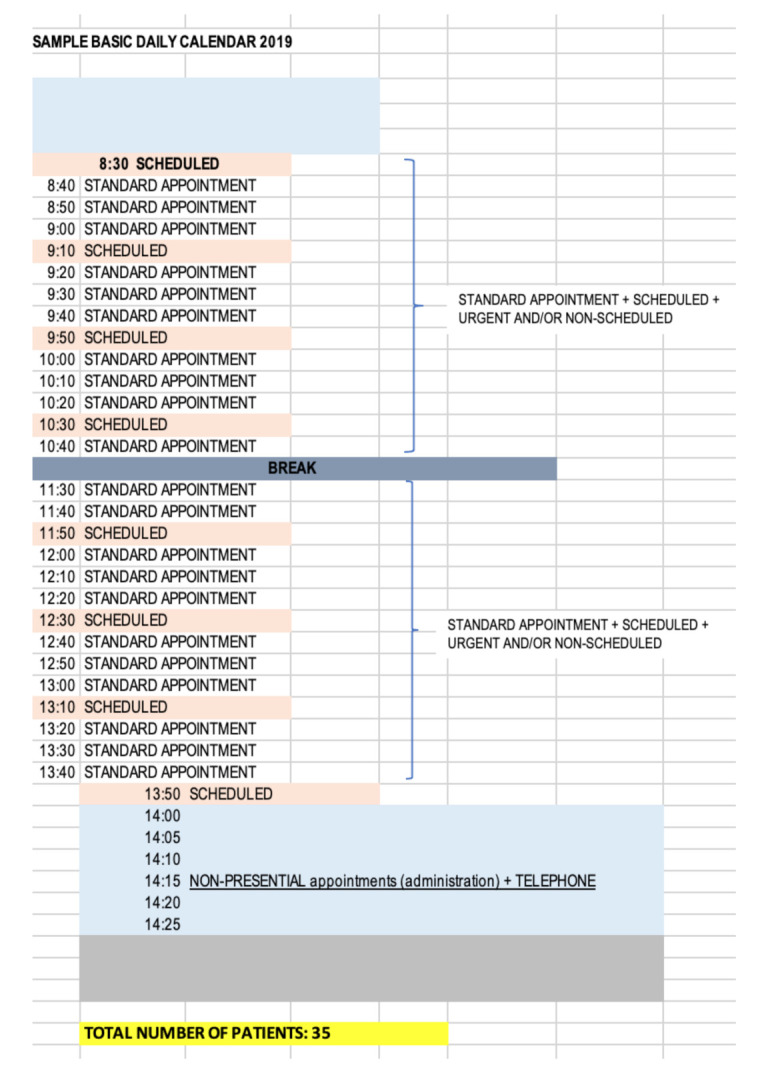
2019 calendar.

**Figure 7 ijerph-18-02876-f007:**
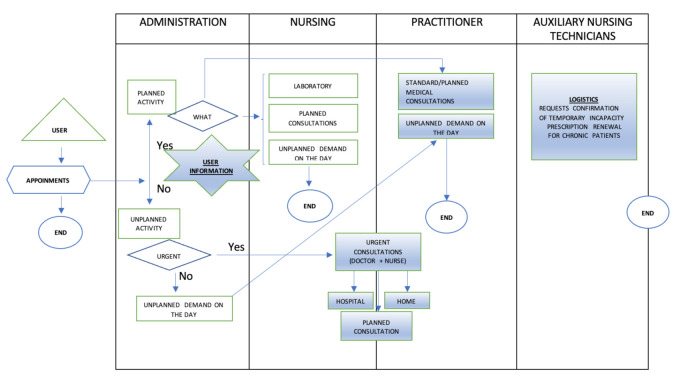
To-Be process diagram.

**Figure 8 ijerph-18-02876-f008:**
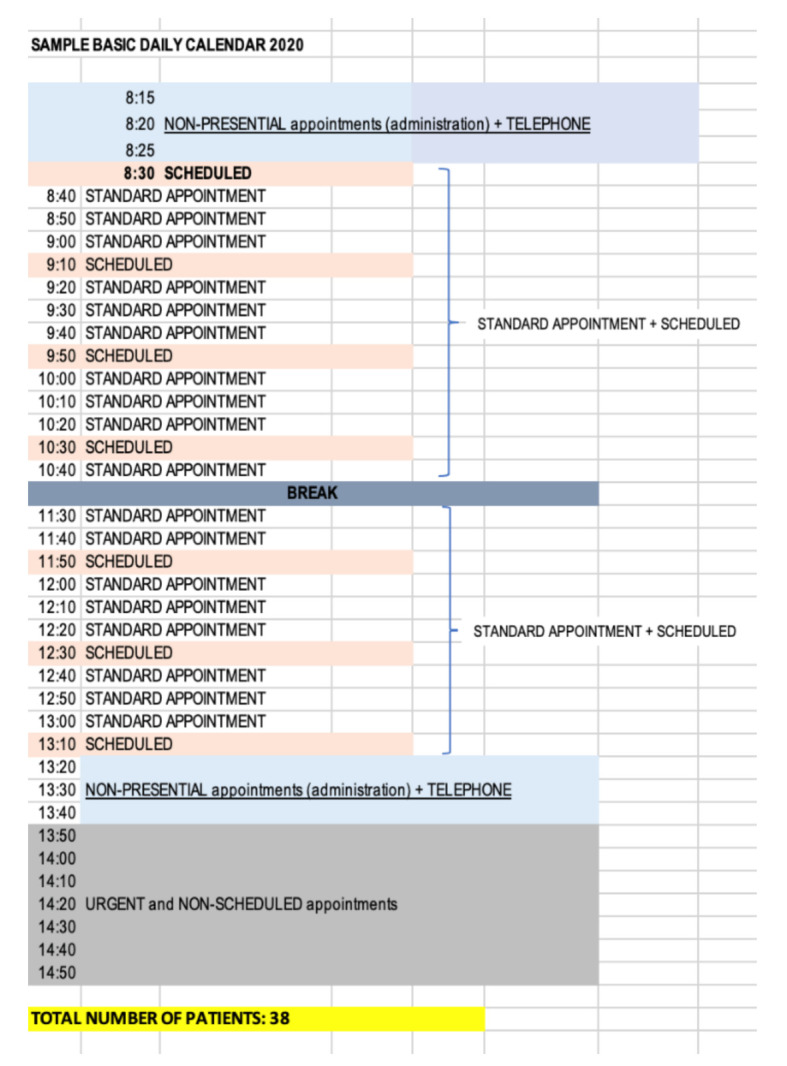
2020 calendar.

**Figure 9 ijerph-18-02876-f009:**
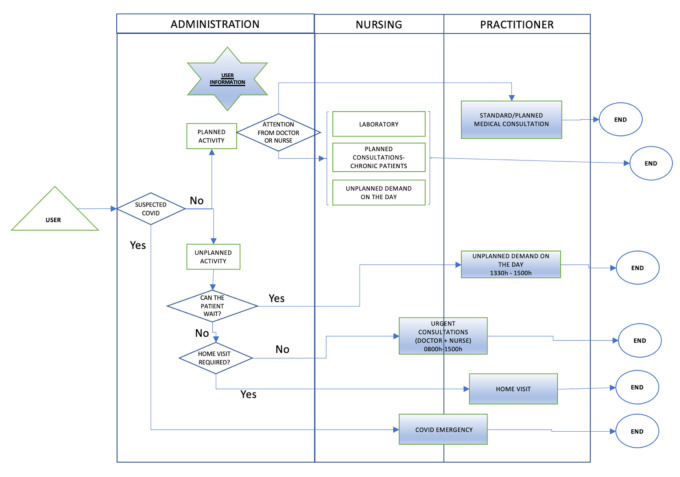
Covid-19 process diagram.

**Figure 10 ijerph-18-02876-f010:**
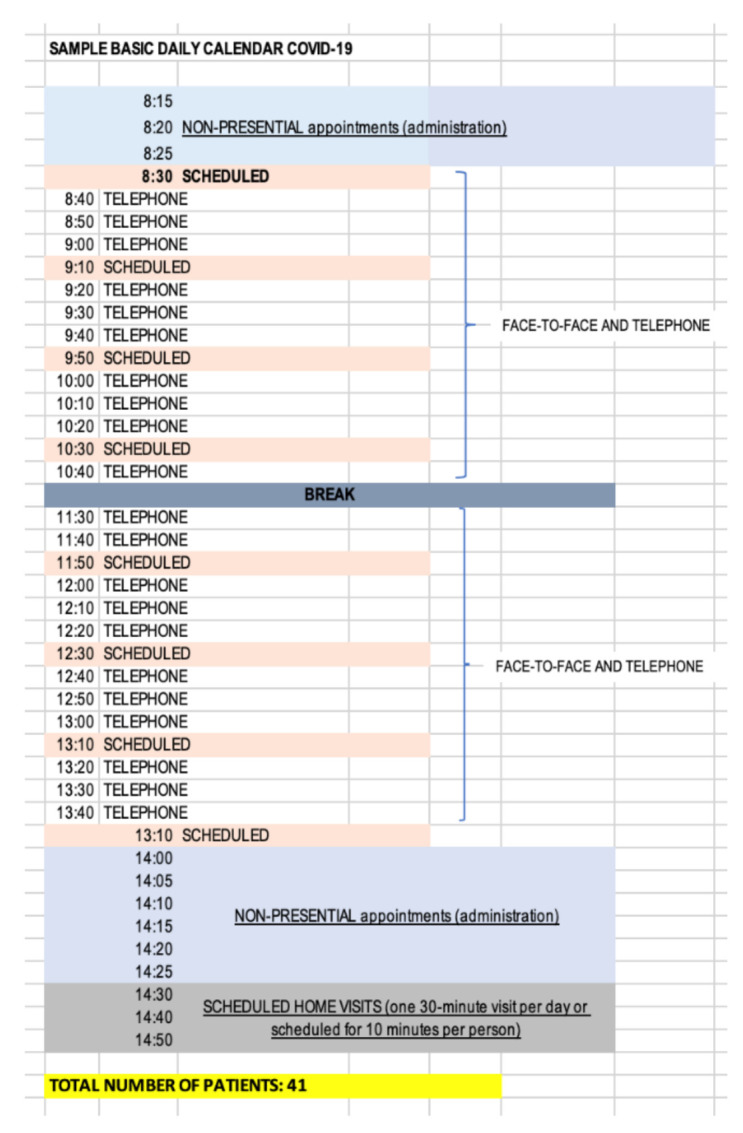
Covid-19 period calendar.

**Table 1 ijerph-18-02876-t001:** Limitations of the Spanish health system.

Area	Problems/Limitations
Primary Care	Closure of health centers. Overwhelmed teleassistance. Lack of diagnostic capacity. Lack of/scarcity of personal protective equipment (PPE). Healthcare overload due to the management of: ○ Suspicious patients. ○ Positives with mild symptoms but confined to their homes. ○ Positives with mild symptoms discharged from hospital emergency rooms. ○ Follow-up of cured patients discharged from hospitals. ○ Referral of primary care professionals to other care facilities. Continuity of cross-team care. Sidelining of control and monitoring of chronic pathologies, such as diabetes, hypertension, etc. Abandonment of primary care-essential disease prevention and health promotion. Out-of-hospital emergency system: ○ Call system saturation. ○ Response delays due to large and expanding transport needs from home and between hospitals. ○ Lack of diagnostic capacity.
Hospitals	Delays in the initial response due to initial public health-controlled reaction restricting testing to those patients who met the probable case conditions, i.e., imported cases. Slow implementation of preparedness programs designed after the H1N1 pandemic. Lack of/poor interoperability (technical, semantic, organizational) between levels of care, hospitals, and regional health systems. No guarantee of the supply of key goods (monitoring devices, gases, respirators, PPE). Lack of/scarcity of PPE. In the socio-health care residences: ○ Lack of/shortage of adequate plans, information, resources, and professionals (until very late). ○ Poor coordination with the NHS.

**Table 2 ijerph-18-02876-t002:** Team and roles.

Post	Role
Director of primary medical care	Improvement promoter (non-participating)
Director of basic health zone	Promoter and facilitator. Team leader
Coordinator of nurses of basic health zone	Support for the facilitator and participant
Primary care team administration and auxiliary positions	Passive subjects: Aware of the process and propose improvements

**Table 3 ijerph-18-02876-t003:** 2019 measurement indicators.

Evolution of Number of Visits	Numerical Value	% (of Total Visits)
Number of daily emergency visits	15,468	23.23%
Number of scheduled visits plus regular appointments	40,935	61.46%
Number of telephone appointments for emergencies	2641	3.96%
Number of telephone appointments for scheduled activities	1890	2.84%
Number of emergency visits of the day scheduled in activity	5670	8.52%
Total visits	66,604	100%
Appointment delay	2.46	

**Table 4 ijerph-18-02876-t004:** Direct care mode.

	Definition	Functions/Tasks
**Upon patient request:** - Ordinary demand appointments - Unscheduled non-urgent - Unscheduled urgent	Daily demand from patients who have requested it by appointment through the App, Web, or at the Health Center	Patient-generated demands (chronic and non-chronic) while integrating part of the program activities and protocols, as well as relevant preventative activities, into these consultations
**Programmed:**	Claims derived from the activity of health personnel or social workers and scheduled by the doctor themselves	Complex processes of chronic patient care or with social vulnerabilities which cannot be developed in the on-demand consultation. In this type of consultation, care and screening programs defined as quality standards to improve the health of the population are also developed. This planned activity is carried out in conjunction with nursing
**Appointment not in person or by phone**	Generated by the doctor themselves	Procedures derived from medical records
**Diagnostic support**	Generated by nursing	Blood extraction, biological sample taking, electrocardiograms, spirometry, wax extractions, blood pressure measurement, Mantoux test, cytology sample extraction, anticoagulation control.

**Table 5 ijerph-18-02876-t005:** Action plan and activities.

Actions	Activities
Reorganize agenda management	A specific calendar was created for daily emergencies to which one human resource (optional) of the 10 existing in the center was assigned as well as a material resource, office specific. Figure 8 below shows this calendar had two modalities of patient care within the same calendar, one for ordinary plus programmed and another for non-urgent unscheduled demands. This modality of attention was distributed in two differentiated time slots from 0800 to 1330 and from 1350 to 1500, this last time slot being where non-urgent unscheduled visits were attended. In this model, the time period for doctors to carry out administrative tasks was from 0815 to 0825 and from 1320 to 1340. Another important action was to separate the circuits of attention to the scheduled visit and the unscheduled visit once the changes were made at the organizational and functional levels of the center.
Reduce bureaucracy	In order to carry out this action, the plan determined which non-healthcare bureaucratic aspects and the urgent daily demand could be eliminated, as they imply an increase in the healthcare burden and could be carried out at alternate levels. The following five activities were carried out: Improving the chronic treatment dispensing circuits. Emphasis was placed on the need for good management and health education for the dispensing of this type of medication. Eliminate the health acts not included in the health benefits regime of the social security system. In order to inform patients, a poster was created that clearly defined the portfolio of services available at the center. Delegation of the dispensation of temporary disabilities discharge was made at the first consultation of the physician, with or without a scheduled appointment. With respect to discharges, future discharges would be prepared for both scheduled and unscheduled visits as long as they were predictable so they could be picked up at the reception desk without the need for a new scheduled medical visit, thus freeing up the time slots for care. If they were discharged at the patient’s request, they would not require an appointment either. Delegation of the printing of temporary disability confirmation reports, whenever possible and when no medical evaluation was required, to the service of the administration and/or Auxiliary nursing technician Eliminate bureaucratic activities in the scheduled consultations: Do not carry out treatments prescribed by the specialist. Every doctor was responsible for their prescription. The primary care physician would not make appointments for annual check-ups of specialists. Each specialist was responsible for their annual check-ups. Do not make referrals led by medical specialists: Do not make complementary requests derived by the specialist to complete their studies. Redirect the consultation to the corresponding service by the receiving service. Do not generate official medical reports.
Improve consultation efficiency	Create internal pathways and redirect the demand towards programmed visits, and establishing protocols of action. The objective was to improve the quality of care of chronic diseases through the following six activities: Promote training activities in the center with the aim of improving pathways of care. To this end, every week the primary care team would meet for one hour to discuss issues related to the levels of care in the center, as well as to update the unified clinical guidelines (which would also improve action 3). Organize the programming of assistance activities, infirmary activities, control of Acenocoumarol, cures, injectables, residencies, consultations of cardiovascular irrigation, etc. Do not attend within the “unscheduled demand” the foreseeable acts such as chronic medication and confirmations of parties. They would only be carried out during the programmed visit and whenever it was not justified by the administrative staff and/or Auxiliary nursing technician. Schedule the complementary tests and/or non-urgent referrals required by the patients according to the clinical guidelines for the control of their disease in the scheduled visits. Improve the quality of pharmaceutical prescriptions, which required the correlation of each clinical diagnosis with its appropriate treatment in the module of pharmaceutical prescriptions. Improve efficiency in consultations through other guidelines: Give long guidelines for chronic treatments. Request “opportunistic” complementary tests. Do not ask for tests outside the protocol. Improve health education in scheduled consultations.
Increase decision making roles by group	Training activities were required to promote leadership attitudes and teamwork. Following are two examples: Administrative Role: Increased capacity to make decisions and appointments on non-urgent pathology calendars during the day and to inform patients of the existing circuits in the Health Center and guide them according to demand. Auxiliary Nursing Technician Role: Increased decision-making capacity in logistics, stock control, dispensing of temporary disability confirmation reports, control of accessibility to the center, and collection and packaging of biological samples.
Keeping patients informed	A dissemination and communication plan was established through the installation of informative posters and notes at the counters of the Health Center with the aim of providing information on the different care pathways (see Reduce Bureaucracy Action 1 above). Community activities were promoted such as educational talks to the patients allowing the differentiation of the word “emergencies” (vital cases) and using the term “unscheduled activity” to make the patient aware of the correct use of the new work processes of the Health Center

**Table 6 ijerph-18-02876-t006:** 2020 measurement indicators.

Evolution of Number of Visits	Numerical Value	% (of Total Visits)
Number of daily emergency visits	9867	15.53%
Number of scheduled visits plus regular appointments	51,225	80.60%
Number of telephone appointments for emergencies	536	0.85%
Number of telephone appointments for scheduled activities	895	1.41%
Number of emergency visits of the day scheduled in activity	1040	1.7%
Total visits	63,563	100%
Appointment delay	2.02	

**Table 7 ijerph-18-02876-t007:** Action plan and achievements.

Actions	Achievements
Reorganize agenda management	This allows physicians to better organize their time as they reorganize their schedules (Figure 8). With these changes, the doctors on the calendar could attend 38 visits with a time of availability of 10 min. In the time range from 0800 to 1330 the visits were for ordinary plus scheduled so the doctor could manage, depending on the requirements and needs of the patient and the type of visit, an average of 10 min availability as they were aware of what they were going to attend, thus reducing stress, uncertainty, the burden of care, and they could ultimately improve patient care. By establishing a time slot (1350 to 1500) for non-urgent unscheduled visits, these did not have to be entered into the calendar at random or to distort the day-to-day planning such so scheduled appointments were not delayed. In addition, 80% of the attention was provided in person and 20% by telematic means. On the one hand, patients who demanded unscheduled urgent activity during the day would be directed to a specific location in the center where vital emergencies were attended with less risk, higher efficiency, and multidisciplinary support, i.e., an extra consultation would be established with personal care resources thus minimizing the patient risk. It also reduced then number of unnecessary trips within the center. On the other hand, patients who demanded a non-scheduled, non-urgent appointment would be scheduled during the day in care modules of the daily work calendar designed for this purpose and according to their demand. This allowed release of the care burden in the time slots of patients who had requested or arranged an appointment in advance, thus increasing the time by scheduled demand and ultimately patient satisfaction including decreasing waiting time.
Reduce bureaucracy	Allowed an increase to the time allowed to attend the patient in their scheduled appointment. Promoted a reduction in work stress due to administrative activity time allocation to the scheduled activity. Allowed for consultation planning and organization.
Improve consultation efficiency	Reduced the care overload due to the protocol familiarity. Reduced stress when dealing with patients. Increased the time allowed to care for the patient in their scheduled appointment. Improved communication with co-workers. Improved patient–physician relationship and trust. Improved patient satisfaction. Allowed early diagnosis and problem solving.
Increase decision making roles by group	Improved the communication with the public. Improved the accessibility and the patient traffic within the center. Improved the agenda management in the programmed activity.
Keeping patients informed	Improved the education of the population. Improved the operation of the center.

**Table 8 ijerph-18-02876-t008:** Action plan and activities—Covid-19.

Actions	Achievements
Reorganize the calendars	The physician who only attended to emergencies was maintained, but they will try to resolve the calendars by telephone. Two physicians, a doctor and a nurse, were added to cover the Covid-19 calendar. In this way, the calendar of the rest of the physicians changed to the structure shown in Figure 10 below. The structure of the calendar during the Covid-19 period for physicians who did not perform Covid-19 assistance (Figure 10) was reorganized as follows: 0800 to 0830 and 1330 to 1425. The physicians performed administrative tasks on medical records. 0830 to 1330. The physicians in the ordinary plus scheduled calendar could attend 41 visits with a time availability of 9.5 min. Here there was a radical change due to capacity restrictions to limit contact. It was decided that ordinary appointments would be used for telematic care (80%) and scheduled appointments would be used to continue face-to-face care for chronic pathologies (20%). 1425 to 1500. This period was intended for one scheduled home visit.
Increase telemedicine (NEW ACTION)	The way of working was changed by promoting “telemedicine”, which was the safest method of care, although not always ideal. To promote and improve telemedicine, an attempt was made to provide more technological resources, but it was not possible due to all the economic resources being earmarked for purchasing PPE and other resources to increase safety in the center. To solve this problem, the existing technological resources were redistributed and interconnections with this department were promoted.
Improve information flow between levels (NEW ACTION)	The process of transferring information between the different levels of the health department was precarious and reliant on manual processes. In many cases, it could take several calls to ensure that all the information transferred between levels. To guarantee continuity of care between the different levels of care, clear referral and fresh communication protocols were established. These protocols were provided with the necessary resources to maintain safe and agile referrals (e-mail, telephone numbers, designation of referrers, clear referral criteria).
Reduce bureaucracy	The process of de-bureaucratization in primary care should be continued in order to eliminate the non-care burden. The situation allowed for continuation of those primary care physician de-bureaucratization activities that were not implemented in the first quarter of 2020 due to excessive pushback including the following: Failure to provide proof of attendance. No sick leave for temporary disability during hospital emergency visits. Failure to carry out complementary test evaluations derived from hospital specialists. Do not make requests for medical transport derived from specialists. The service itself will be responsible for issuing requests to the patient for future appointments. Do not carry out analyses derived from specialists. The specialist themselves will be in charge of scheduling them.
Increase decision making roles by group and delegate more responsibilities and functions	The needs in the current situation demanded that all personnel become much more decisive, not only because efficiency gains, but also because it was safer for physicians, professionals, and patients. In order to be more decisive and to eliminate the workload on the staff, the competencies and functions of the nursing staff were extended to perform previous triage of patients for clear cases, for which they were perfectly qualified.

**Table 9 ijerph-18-02876-t009:** Covid-19 period indicators.

Indicator	Result
Number of telephone appointments in scheduled activity	36,648
Number of face-to-face appointments in scheduled activity	9665
Number of unscheduled urgent appointments	1800
Number of appointments in Covid-19 calendar	4950
Total visits	53,063
Delay	4.78

**Table 10 ijerph-18-02876-t010:** Action plan and improvements—Covid-19.

Actions	Improvements
Reorganize the calendars	This continued to allow the physician to manage according to the needs and requirements of the patient and the type of visit, as they were aware of what they were going to attend. In the same way as in the previous period, stress, uncertainty, and the burden of care were reduced and patient care was improved. In addition, it allowed for efficient management of the work agenda.
Increase telemedicine (new action)	These measures made it possible to: Increase the accessibility of the population through on-line platforms. Diagnose pathologies such as dermatological pathologies more quickly and fluently and their prompt referral to the specialist. Speed up the sending via e-mail of documents necessary for patients’ administrative procedures: temporary incapacity reports, supporting documents, medical certificates, diagnostic tests. Improve the doctor–patient relationship and communication.
Improve information flow between levels (new action)	The establishment of new care communications between the different levels of care (primary-hospital) made it possible to: Increase the capacity for early diagnosis. Increase problem-solving capacity with the different specialties. Increase decision-making capacity. Increase capacity of preferential guided referrals of acute pathology, decreased patient morbimortality. Centralize care needs in departmental protocols. Strengthen primary care as the central axis of health care.
Reduce bureaucracy	Continued de-bureaucratization helped to improve the administrative burden on the center’s staff so that they could provide better service and care to patients. This improvement involved increasing the dedication of the practice to purely clinical matters, decreasing the care load and improving the quality of work.
Increase decision making roles by group and delegate more responsibilities and functions	The expansion of functions and competencies allowed: Ability to control entries to the medical center and their corresponding referrals. Improvement in the effectiveness of triage. Improvement in the resolution of bureaucratic problems. Enhancement of the improvements already achieved.

**Table 11 ijerph-18-02876-t011:** Summary of comparative situations 2019–2020.

	Q4 2019	Q1 2020	Covid-19
Times	Average care for 50 patients - Patient care time: 11 min. - Actual service time: < 7.8 min - Average delay: 2.46 days	Average care for 38 patients - Patient care time: 10 min. - Approximate average actual service time distributed according to requirements: 10 min - Average delay: 2.02 days	Average care for 41 patients - Patient care time: 9.5 min. - Approximate average actual service time distributed according to requirements: 9.5 min - Average delay: 4.78 days
Physicians	- Inability to plan the calendar. - Inability to manage time. - Stress. - Overload.	- Ability to plan the calendar, separating scheduled and non-urgent calendar from the calendar of the daily emergencies. - Improved time management. - Reduced stress of uncertainty about the time of emergencies incoming and their effect on scheduled visits. - Reduced overload.	- Undeniable period of work stress caused by high uncertainty in how to work. This reduced with new schedule implementation, note the constant changes in planning and number of contagions may pose new challenges of care overload that are difficult to plan for. - Dedicating resources to Covid-19 reduced the initial overload and chaos. - Improved working culture introduced.
Nursing	- Remote consultations. - Physician assignment time for emergencies. - Work stress. - Anxiety fueled by uncertainty.	- Remote consultations. Designed to establish shared medical-nursing spaces. - Physician assignment time for emergencies. There was a dedicated doctor for emergencies. - Work stress reduction due to protocol awareness and knowing what to do and how to do it. - Anxiety reduction when faced by uncertainty due to protocol awareness and knowing what to do and how to do it.	- New reorganization and expansion of roles to adapt and provide quality patient service. - Work stress returned until a new way of working was introduced.
Administration	- Overcrowding due to unscheduled daytime emergency visits. - Work-related stress due to the perception of poor patient care.	- Saturation due to daily unscheduled emergency visits reduced by introducing the emergency calendar of the day with a dedicated physician for the calendar and to support the other physicians. - Work stress due to the perception of poor patient care by the reduced ability to deliver improved care.	- New reorganization and expansion of roles to adapt and to provide quality patient service. - Work stress returned until a new way of working was introduced.
Patients	- Little attention from the physician. - Loss of time.	- Better physician care introduced. - Loss of time reduced by in-place protocols and elimination of unnecessary internal displacements.	- Period of uncertainty and fear. - Uncertainty about how to request visits (need for information and training).
General	- Lack of protocols. - 80% face-to-face service and 20% telematic service.	- New protocols launched. - 80% face-to-face service and 20% telematic service maintained.	- Difficulty in establishing protocols, though ultimately carried out. - 20% face-to-face care (this allowed social sustainability of chronic pathologies) and 80% telematic care.

## Data Availability

The data presented in this study are available on request from the corresponding author.

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
