# Peer review of "A Model for the Implementation of Lean Improvements in Healthcare Environments as Applied in a Primary Care Center"

_ijerph, 2021, doi:10.3390/ijerph18062876_

Round 1
Reviewer 1 Report
Line 9: Change "... Lean Manufacturing is a proven philosophy in ..." to "... Lean Manufacturing is a proven philosophy in ..." - this is because "Lean Manufacturing" is the line applied to manufacturing. When lean is applied to the health sector, it used to be called "lean healthcare". I think the authors here want to refer to the beginning of lean philosophy only;
Line 19: Change "Lean Manufacturing" to "Lean Philosophy"
Line 92: Change "... Lean Manufacturing philosophy ..." to "... Lean philosophy ..." - remove "Manufacturing" as it does not make sense;
Line 119: Change "... Lean Manufacturing was born in the industrial sector ..." to "... Lean Manufacturing was born in the industrial sector ...";
Line 145 and 146: There is great confusion here with the lean tools on the part of the authors. The TPM has nothing to do with the "Team approach to problem solving". The TPM is just "Total Productive Maintenance"! The "Team approach to problem solving" as a tool could be, for example, A3 Problem Solving. The TPM is geared specifically towards maintenance;
Line 589: The authors could include here other tools such as WID (Waste Identification Diagrams;
Line 1111: Clarify ... When the authors refer to "tools", what tools are they referring to? Lean tools? If it’s lean tools, you should sort and write "... lean tools ...";
Line 1578: Why did the authors in "Limitations and future lines of research", not recommend for future research, consolidate the study with the introduction of Six Sigma? In other words, through Lean Six Sigma?
Author Response
Please see the attachment." in the box if you only upload an attachment

Reviewer 2 Report
The authors have significantly improved the document. The document is now well structured and easy to read. The stages proposed in the methodology can be analysed independently and without confusion.
I think the paper is now publishable in the journal. However, I have a suggestion for the authors.
In lines 786 to 790 the authors textually write:
“In order to do this the most appropriate tool had to be chosen. As discussed above one of the most appropriate tools was considered to be VSM. However, due to the lack of team training and in order to introduce the team to this type of tool, it was decided best to start with a process diagram, as shown in Figure 4 below. Meanwhile a training plan was being established for the team that would allow them to perform VSM in the future”
In the introduction, the authors repeatedly mention the need to use VSM as the first step of the Lean methodology. However, they then go on to say that they cannot use this lean tool because of a lack of training for the researchers.
I propose that:
- In the introduction describe the alternative tools to VSM, with their advantages and disadvantages compared to VSM.
- The authors justify the use of process diagrams and the lack of training of researchers in VSM is not mentioned.
- Authors should delete the above-mentioned paragraph.
Author Response
Please see the attachment

This manuscript is a resubmission of an earlier submission. The following is a list of the peer review reports and author responses from that submission.
Round 1
Reviewer 1 Report
- The authors in the abstract say that lean is a methodology, which it is not. For all the effect in the scientific community lean is considered as a philosophy, which is implemented in organizations through practices and tools (lean tools).
- (line 55) The author García is referred to in the references as being the 2nd or the 3rd, however consulting the final references to the 2nd and the 3rd, this author does not appear. The reference to García, appears further down in other references but not in these.
- (line 83) The authors mention that the most used lean tool is VSM, which I have many doubts. How do the authors make this statement? Did they do any research or an extensive literature review? I have many doubts about this statement. From my research I have concluded that the most used lean tool by organizations is the "5S"... Authors should review this information!
- (line 111) Why do authors put the pages in some references? Any special reason? They should review the references. In this same line the reference [28] associated to the "IMAS" still appears, but consulting the references at the end of the article the reference [28] has as authors "Hernández, J. & Vizán, A."... what happened??
- The authors mix "lean tools" with "quality tools", and say they are all "lean tools"... some confusion... For example, Ishikawa diagram is a "lean tool" or a "quality tool"?? (line 190). The 5 Why, is it a "lean tool" or a "quality tool"??? (line 213).
Reviewer 2 Report
Authors show an interesting study on improvements in primary care efficiency using Lean manufacturing tools.
The aim of the study is very ambitious but the paper needs to be improved.
The presentation of the document is chaotic.
- Line numbering to the left of the document.
- Bullet list with -, •, ✓, numbers. This is not homogeneous.
- The development of Phase 6 (line 539 to 666) impedes their understanding. It is totally unmanageable for the review.
- Et cetera.
I think it is a very good research presented horribly.
I suggest that detail be given to the organisation of the document to facilitate the reading and interpretation of the results obtained in the three scenarios presented.
If the document is structured in a coherent way, the final result will be of the highest scientific quality.